# Policies, Guidelines, and Practices Supporting Women’s Menstruation, Menstrual Disorders and Menopause at Work: A Critical Global Scoping Review

**DOI:** 10.3390/healthcare11222945

**Published:** 2023-11-10

**Authors:** Danielle Howe, Sarah Duffy, Michelle O’Shea, Alex Hawkey, Jon Wardle, Sophia Gerontakos, Linda Steele, Emilee Gilbert, Lara Owen, Donna Ciccia, Emma Cox, Rebecca Redmond, Mike Armour

**Affiliations:** 1NICM Health Research Institute, Western Sydney University, Sydney 2145, Australia; d.howe2@westernsydney.edu.au (D.H.);; 2School of Business, Western Sydney University, Sydney 2751, Australia; 3Translational Research Institute (THRI), School of Medicine, Western Sydney University, Sydney 2145, Australia; 4National Centre for Naturopathic Medicine, Faculty of Health, Southern Cross University, Lismore 2480, Australia; 5Law Health Justice Research Centre, Faculty of Law, University of Technology Sydney, Sydney 2007, Australia; linda.steele@uts.edu.au; 6School of Psychology, Western Sydney University, Sydney 2751, Australia; 7School of Modern Languages, University of St Andrews, St. Andrews KY16 9PH, UK; 8Endometriosis Australia, Sydney 2000, Australia; 9Endometriosis UK, London SE1 1SZ, UK; 10Medical Research Institute of New Zealand (MRINZ), Wellington 6242, New Zealand

**Keywords:** menstruation, menstrual disorders, menopause, workplace, employer guidelines, organisational policy

## Abstract

(1) Objectives: This paper presents a scoping review of global evidence relating to interventions (i.e., policies, practices, guidelines, and legislation) aimed at supporting women to manage menstruation, menstrual disorders, and menopause at work. (2) Methods: Databases including Medline (Ebsco), CINAHL (Ebsco), Scopus, Web of Science, APA PsychInfo (Ebsco), Humanities International Complete (Ebsco), Academic Search Premier (Ebsco), HeinOnline and OSH Update, and Google Scholar were searched in May 2022. (3) Results: Of 1181 unique articles screened, 66 articles are included. Less half of the articles (42%, 28/66) presented/reviewed an intervention related to women’s workplace health. A total of 55 out of the 66 articles are set across 13 countries with the remaining 12 articles described as multi-country studies or reviews. Half of the articles presenting/reviewing an intervention were grey literature, with several undertaken in UK and EU member countries. Interventions focusing on supporting women with menopause at work were the most common (43%, 12/28), followed by menstruation (25%, 7/28) and menstrual disorders (7%, 2/28). Across the reviewed articles, recommendations were categorised as adjustments to the physical work environment, information and training needs, and policy and processes. Few articles explicitly presented or affirmed a design-process and/or evaluation tied to their intervention. In lieu of design-process, this review categorises the rationales driving the development of an intervention as: pronatalist, economic rationalism, gendered occupational health concern, cultural shift towards gender equity objectives, and efforts to reduced shame and stigma. (4) Conclusions: There is a growing body of evidence aimed at understanding women’s experiences of managing their menstrual and reproductive health in the workplace and how this impacts their work/career trajectories. However, little research is explicitly concerned with exploring or understanding interventions, including their design or evaluation. Most articles report menopause guidelines and are typically confined to the UK and EU-member countries. Despite the prevalence of menstrual disorders (e.g., endometriosis and polycystic ovarian syndrome (PCOS)) there is limited literature focused on how women might be supported to manage symptoms associated with these conditions at work. Accordingly, future policies should consider how women can be better supported to manage menstruation and menstrual disorders at work and recognise the importance of co-design during policy development and post-intervention evaluation. Further research needs to be undertaken on the impact of workplace policies on both employers and employees.

## 1. Introduction

There is increasing interest in policies, practices and interventions that support women (The authors of this scoping review recognise that not all people who experience menstruation, menopause, and/or menstrual cycle disorders identify as women. Trans men, intersex, non-binary, and gender diverse people also experience menstruation, menopause and/or disorders of the menstrual cycle; and face similar and unique challenges to cis women. This paper uses the term ‘women’ to acknowledge there is a gendered element to managing reproductive health while at work. We chose the term ‘women’ to reflect the large number of articles in the scoping review which focused on cis women’s experiences, and therefore to not erase or oversimplify the unique challenges faced by trans men, intersex, non-binary, and gender diverse people who are also managing their reproductive health in the workplace. Through this generalist approach, we highlight the gap in literature regarding the experiences of trans men, intersex, non-binary, and gender diverse people who are also managing their reproductive health in the workplace and hope this research can be built upon in the future.) in managing menstruation, menstrual cycle disorders, and menopause at work [1,2]. In the early twentieth century, the Soviet Union and Japan implemented national menstrual leave legislation. However, these interventions were underpinned by pronatalist sentiment tied to the erroneous belief that strenuous work impinged on women’s capacity to produce offspring, limiting women’s role to mothers only [1]. Recently, private organisations have implemented menstrual and/or menopause policies, including flextime and/or leave entitlements as part of “progressive” human resource management policies [1,3]. Other stakeholders, including trade unions, have leveraged occupational health and safety legislation to create guidelines that may include awareness training and managerial education on the topic [2]. 

Women have nearly proportionate representation in the workforce across Australia, North America, United Kingdom (UK) and Europe [4,5]. Consequently, almost half of all workers are likely to manage symptoms associated with menstruation and/or menopause [6,7,8]. Gynaecological and menstrual disorders such as dysmenorrhea (period pain that occurs without any physical changes in the pelvis affecting up to 90% of women and those assigned female at birth (AFAB)), endometriosis (a chronic inflammatory disease affecting ~10% of women and AFAB (The authors recognise how binary gendered categorisations are misleading and inaccurate when tied to endometriosis research reporting. To rectify these inaccuracies, we have included the term: ‘assigned female at birth’ (AFAB).)), and PCOS (a chronic metabolic and reproductive disease affecting between 4–22% of women and AFAB globally) significantly impacts work attendance and workplace productivity [6,9,10,11,12]. For example, endometriosis is a chronic and painful clinical condition that can impact on women’s physical, emotional and economic well-being [13]. Fatigue, severe pain, and the side effects of treatment can lead to significant periods of sick leave or being unfit for work, resulting in forced part-time work or cessation of employment [14]. 

The cost of endometriosis to the Australian economy is an estimated $9.7B (AUD) annually. Most costs (84%) relate to a loss of productivity [15], with similar impacts seen globally [16]. The prevalence of menstrual disorders and their significant individual health and economic impacts emphasize the need to develop and implement effective interventions to support the affected workforce [7]. For instance, flexible working arrangements were found to benefit endometriosis symptom management [17]. In one Australian study, the majority (79%) of respondents reported how endometriosis symptom management was easier when working from home, while productivity was enhanced [17], highlighting the positive physical and economic impacts of hybrid/flexible work policies. 

While three systematic reviews have focused on experiences of work and menopause [2,18,19] and one recent review on human resource (HR) policies and legislative menstrual leave [1], no prior review has explored interventions across the menstrual lifecycle. Given the lack of consolidated information on what workplace policies/guidelines or interventions are available to women across their reproductive lifespan to support common menstrual/gynaecological conditions such as endometriosis, this scoping review aimed to map the types of interventions available for different stages of the reproductive lifecycle, the rationale(s) driving their design, how they were implemented, and how the outcome of the intervention was evaluated. This information can be used to identify potential areas of unmet need, gain a better understanding of how policies, guidelines or interventions have been designed and/or implemented, and the impact of these changes for employers and/or employees

### 1.1. Aim

To provide a critical scoping review of interventions supporting women managing menstruation, menstrual disorders, and menopause in the workplace; categorise and summarise data presented in relation to interventions in order to increase our understanding of the current literature and identify research gaps.

### 1.2. Objectives

This review:maps evidence from workplace interventions specific to menstruation/menstrual disorders/menopause at work;categorises data presented in interventions, (type/rationales/recommendations); andsynthesises learnings from implementation and evaluation.

## 2. Methods

### 2.1. Review Design

A scoping review is appropriate to provide an overview of the evidence, identify and clarify key concepts and outline any research gaps [20]. In contrast to a systematic review, scoping reviews present a broad overview of the evidence pertaining to a topic, irrespective of study quality, and can be used for hypothesis generation to inform future work [21]. This review accords with the JBI methodology and the Preferred Reporting Items for Systematic Reviews and Meta-Analyses extension for Scoping Reviews (PRISMA-ScR) [22,23,24].

### 2.2. Review Question

What knowledge exists regarding interventions designed to address menstruation, menstrual disorders, and/or menopause at work? 

### 2.3. Search Strategy

On 14 May 2022 ten electronic databases were searched (with no date restriction): Medline (Ebsco); CINAHL (Ebsco); Scopus; Web of Science; APA PsychInfo (Ebsco); Humanities International Complete (Ebsco); Academic Search Premier (Ebsco); HeinOnline and OSH Update; and Google Scholar. When searching Medline and CINAHL, MeSH terms were used where indicated. Additionally, ‘MM’ searches the exact subject heading in APA; ‘DE’ refers to ‘heading’ or ‘keyword’ when searching Academic Search Premier and Humanities International Complete. 

We combined subject headings/key words, inclusive of: MeSH/MM/DE workplace, MeSH occupational health, MeSH/MM/DE Policy exp MeSH/MM/DE Jurisprudence, MeSH/MM/DE Menstruation, MeSH/MM/DE Menstruation disturbances exp, MeSH/MM/DE Menopause exp, MeSH/MM/DE Endometriosis, MeSH/MM/DE Polycystic ovary syndrome; and text words/subject terms, such as workp*, works*, guideline*, intervention*, program*, Polic*, Law* Guideline* Program*, Intervention*. The ‘*’ refers to truncated word roots to capture multiple derivations, e.g., menstrua* will capture menstruation, menstruator, menstruating. This strategy was used in Medline, CINAHL, APA PsychInfo, Academic Search Premier, and Humanities International Complete (Table 1) 

Four search strings using a combination of text words/subject terms for Scopus, Web of Science, Heinonline, and OSH Update (Table 2). In addition, after searches were complete, experts in the field were also consulted to identify any uncaptured studies. 

### 2.4. Selection Criteria

A systematic literature search strategy was inclusive of all articles (empirical research, systematic reviews, meta-analyses, conference abstracts, industry reports, and HR or related documents) until the 14 May 2022. Articles/documents were included based on the criteria (in Table 3), as determined by the JBI framework [22].

Non-English items, studies relating to maternity care, pharmaceutical interventions, ‘women’s health’ were outside our inclusion criteria (Table 3). Articles before 1945 were deemed irrelevant. 

### 2.5. Study Screening and Data Charting

One researcher (SG) generated the initial 1579 items. After removing duplicates, 1169 items were screened by title and abstract by two researchers (SG, MA) independently using the software COVIDence, version 2022, excluding 1107 articles. If uncertain of eligibility, articles were advanced to full-text review. Sixty-two reports were sought for retrieval, one was unable to be accessed. 

A third researcher (DH) assessed the eligibility of the remaining 61 texts. A data-charting form was developed to document author(s), study country/setting, target population, study design, type of intervention, rationale driving intervention, evaluation of intervention and key findings related to the article’s objectives. Five articles were excluded due to: being published prior to 1945 (n = 1); relating to pharmaceutical intervention (n = 1), outside the context of the workplace (n = 2) and broadly to ‘women’s health’ (n = 1). 

Twelve additional studies were identified through other methods including: expert consultation (described below) (n = 5), additional citations found through reference lists of included articles (n = 2), and conferences abstracts (n = 4). DH determined eligibility using the data-charting form and excluded two articles for irrelevance. The remaining met eligibility and were included in the review *(*Figure 1). 

### 2.6. Expert Consultation and Contribution

To enhance the reviews’ rigour, after the initial data extraction, the findings (including the list of included studies) were shared with eight experts from fields across disability, psychological equity and diversity, menstrual culture, and legislation, education, and clinical research relevant to menstrual health and menstrual disorders. Experts were asked to identify any additional relevant articles that may have not been captured by our searches.

## 3. Results

Out of 1181 articles, 66 items met the inclusion criteria. Study screening and data selection is in accordance with PRISMA 2020 flow diagram (Figure 1) [25]. 

### 3.1. Article Characteristics

The geographical setting of the articles is noteworthy (Table 4). The UK has the largest representation of articles 42% (n = 28), followed by Australia, which has seven articles in the review (11%, n = 7/66).

Articles are categorised into ‘presents or reviews an intervention’ or ‘does not present or review an intervention.’ Twenty-eight articles (42%, 28/66) presented or reviewed an intervention as shown below (Figure 2).

Half of these articles were grey literature (52%, 14/28). 

See Table 5 for the categorization of articles that present or do not present/review an intervention. 

Thirty-eight articles that met the inclusion criteria (58%, 38/66) provided evidence or theory that may contribute to the development, design, or advocacy of an intervention. Most articles (61%, 23/38,) presented primary data that sought to understand women’s experiences managing menstruation at work. This data could contribute to the development of a workplace intervention. For example, Besen et al.’s (2021) work is a qualitative cross-sectional study to understand the effects of menopausal symptoms on work life and could support the development of workplace policies [26]. 

### 3.2. What Is the Focus of the Interventions as They Relate to Women’s Health? 

Seven articles (21%, n = 7/28) described an intervention focused on menstruation. Two articles focused on supporting women with menstrual disorders at work. Safe Work Australia (2019) produced an endometriosis information sheet, including recommendations to help employers support employees with endometriosis. The second article developed guidelines to support women with endometriosis at work as well as menopause [27]. Despite its prevalence and impact, no articles provided interventions to support women with PCOS at work. Interventions focusing on supporting women with menopause at work were the most common (43%, n = 12/28), with half (6/12) sharing guidelines for managers and staff from large organisations/institutional bodies in the UK [28,29,30,31]. 

See Table 6 for comprehensive coverage of the 24 articles that reviewed or presented an intervention. 

### 3.3. What Is the Design-Process for Interventions?

Ten articles (36%, n = 10/28) explicitly stated their design process to develop the intervention. The most common was a participatory approach (n = 5) and synthesis of evidence (n = 4) such as a systematic or narrative review. Eighteen articles (64%, n = 18/28) that presented or reviewed an intervention provided insubstantial information about the design processes employed for an assessment to be made. 

### 3.4. What Is the Rationale Driving the Policy, Guidelines, or Legislation?

Of the articles reviewed, five thematic rationales were identified and labelled as: pronatalism, economic rationalism, gendered occupational health concern, cultural shift, reducing shame and stigma. 

#### 3.4.1. Pronatalist

Pronatalist interventions aim to protect women’s reproductive ability above redressing workplace inequities [1,37]. Three articles [36,37,38] present interventions underpinned by pronatalist rationales from national labour codes in Taiwan and Indonesia, while another reviews pronatalist menstrual policies globally [1]. 

In Indonesia, researchers found the 2003 entitlement to menstrual leave originates in the Republic’s first labour law (1948), which banned women working between 6pm and 6am, and placed limitations on work “deemed to present moral danger to women” [37]. 

#### 3.4.2. Economic Rationalism

Economic rationalism describes interventions that present a business solution to women’s health issues which interfere with productivity, presenteeism and staff retention [1]. Six articles found are underpinned by economic rationalism. Dean (2018) introduced their article by recognizing that nurses transitioning through menopause in the National Health Service (NHS) were less productive than their colleagues. They noted how workplace accommodations improved employee productivity [42]. In association, these articles addressed physiological needs, but failed to offer interventions relevant to training, improved knowledge, and workplace culture. 

#### 3.4.3. Gendered Occupational Health Concern

Integrating gender through occupational health and safety policies was the most common motivation disclosed (39%, n = 11/28). The rationale is that women and men should *both* benefit from occupational health and safety policy and practice. Eleven articles recognise specific physical and psychosocial health risks that only affect women and are insufficiently addressed. It is the responsibility of the union, federation, or employer to mitigate these gendered occupational health risks in accordance with national legislation, such as Australia’s Work Health and Safety (WHS) laws or the UK’s Health and Safety at Work Act [28,30,31,34,40,51]. Interventions included education for workplace leaders (e.g., line managers or union representatives), health and safety checklists, flexible working arrangements, and material accommodations (i.e., access to cold drinking water). 

#### 3.4.4. Cultural Shift to Achieve Gender Equity in the Workplace

Six articles are motivated by a commitment to workplace gender equity. These interventions acknowledge that women are managing both the physiological and symbolic realities of their bodies and in turn are faced with substantive equity issues in the workplace [7]. In short, they are paid less, are less likely to be promoted and are more likely to be harassed [3,41]. These interventions focus on a cultural shift that addresses gender equity issues such as gendered ageism, lack of women in leadership and seek to address the stigma and shame often associated with attitudes relevant to menstruation, menopause, and menstrual disorders [3,41]. 

To achieve a culture shift across industries, Reese et al. (2021) recommends a top-down strategy including: conducting training, developing health and wellbeing policies, or workplace frameworks, to providing tangible accommodations, opening conversations to allow disclosures, zero-tolerance to bullying, and education on how menopause can affect the ability to work, and how working conditions can impact menopausal symptoms [49].

#### 3.4.5. Reducing Shame and Stigma

A multi-organisational review from Bennet et al. (2021) considers three organisations (Australia n = 2, New Zealand n = 1) who developed interventions to reduce menstrual stigma and shame around menstruation. For instance, Australian business Modibodi aimed to normalize internal organization discussions about women’s reproductive health and wellbeing, in part through a paid leave policy for menstruation, menopause, and miscarriage [48]. Relatedly, Modibodi sells period products, highlighting the potential for commercial imperatives influencing this practice.

### 3.5. Recommendations

Recommendations within the interventions presented (28 articles) are labelled (1) physical work environment; (2) information and training needs; and (3) policy and processes (Table 7). 

Half of the articles (50%, n = 14/28) recommend ‘promoting and conducting awareness raising campaigns’ that focus on ‘gendered health’. Just under a third of articles (29%, 8/28) recommended ‘gender-sensitive’ or ‘gendered health and safety checklists’.

Recommendations to alter the ‘physical work environment’ were also common. Nearly one-third of the articles (29%, 8/28) recommended access to cold drinking water and/or food. One-quarter of the articles (25%, 7/28) recommended that work environments have accessible and clean toilets, as well as adequate ventilation and temperature control.

### 3.6. How Were Interventions Implemented and Evaluated?

#### 3.6.1. Implementation

All articles provide recommendations, however, there is no detail on how each organisation or population group should implement the guidance or work collaboratively to implement interventions acro affiliations [3,27,49]. 

#### 3.6.2. Evaluation

Four articles (14%, n = 4/28) (Table 8) provided guidance for the evaluation of the interventions they presented or reviewed. EU-OSHA (2014) presented the most extensive evaluation strategy, which included recommendations to set objectives, obtain rapid feedback from women in the workforce, assessments during periodic meetings with management and union representatives, and provided evaluation tools and checklists that could be used during assessments and post-trainings to understand the usefulness and impact of the mentioned interventions. 

Twenty-four articles (86%, n = 24/28) included no information about how the intervention was evaluated. In some cases, articles reported impact data, however, no evaluation approach was reported. For example, Bennet et al. (2021) reported impact data relevant to the Victorian Women’s Trust menstrual and menopause policy. Following the implementation of the policy, employees were more committed to their workplace, and the workplace was more open to discussing menopause, reducing stigma. Problematically the sample size or evaluation is not disclosed [48]. This identified evaluation gap is also noted by Baird et al. (2021) [1].

#### 3.6.3. Industry

Table 9 summarises all 66 articles included in the scoping review, categorizing them by geographical setting, industry or working environment, study design, and whether the articles present or review an intervention. 

There was significant representation from trade unions, with many articles (42%) from the UK. Twelve articles (18%, n = 12/66) examined as their target workforce trade unions and federations. Eight articles focused on trade unions and federations (12%, n = 8/66) are from the UK. The nursing sector was represented through three review articles (4%, n = 3/66) and is the second most represented workforce.

Articles focusing on trade unions and federations were produced by governing bodies, which each applied the rationale of ‘gendered occupational health concern’ to drive the development and implementation of guidance. Further understanding of the relationship between these variables is outside the scope of this review.

## 4. Discussion

### 4.1. Menopause Advocacy

This review illustrates how existing interventions mostly focus on menopause, with less published work specific to menstruation and menstrual disorders in the workplace. Most articles were set in the UK, with a focus on trade unions/federations. Understanding why there is an extensive literature focused on interventions for menopausal workers from trade unions and federations in the UK may provide learnings for other countries and/or industries. 

We hypothesise that rationale, age, and visible physiological symptoms may contribute to the significant number of articles focused on interventions designed for menopausal workers. Firstly, all reports (from trade unions and federations) framed menopausal policies, recommendations and guidelines, as a ‘gendered occupational health concern.’ That is, men and women have different physical and psychological health risks that need to be addressed by the union, federation, or employer. Gender-sensitive occupational policies, such as menopause guidelines, are positioned as employers’ common law duty of care under the UK’s Health and Safety at Work Act 1974, the Management of Health and Safety at Work Regulations 1992, and the Equality Act 2010 [2,51]. Menopause is relevant to three of the eight protected characteristics (age, gender, and disability) in the Equality Act 2010, which is cited in most of the trade union reports—and may explain the focus on guidelines for menopause at work [2]. Age may account for the focus on menopause guidance in trade union and federation reports. Menopause typically takes place between 45–55 years, an age at which people are more likely to be in senior roles and therefore may have greater influence and access to the networks and resources to implement policies and practices and commission research [26]. The visible physiological symptoms, such as vasomotor symptoms (i.e., hot flushes), are more likely to affect health and safety within industries run by trade unions [43]. Hot flashes may be exacerbated by uniforms, PPE, or high thermal work environments (e.g., commercial kitchens) [46,51]. 

Menstruation and menstrual disorders should be included within gendered occupational health, but there may be several reasons why they are less frequently represented. First, dysmenorrhea, PCOS, or endometriosis symptoms are typically less visible. Second, societal pressures to ‘hide’ menstruation may encourage the concealment of problematic menstrual signs or symptoms [85,86]. Additionally, menopause may be perceived by employers as more manageable to support in terms of anticipated time off work, as it lasts fewer years than menstruation. Further investigation of the relationship between legislation and workplace policy may be beneficial to wider implementation. Using national legislation, like the UK’s the Management of Health and Safety at Work Regulations 1992 and the Equality Act 2010, could provide a framework for further interventions motivated by gendered occupational health. 

The presence or absence of an appetite for change and inclusion in education and training are an important consideration. Safe Work Australia (2019) published a set of broad recommendations on how a workplace could support individuals with endometriosis. The report highlights the legal obligations by employers to make reasonable workplace adjustments with respect to the potential disability caused by endometriosis [34]. However, an Australian union, the Health and Community Services Union (HACSU), introduced a broader reproductive health and wellbeing policy [48]. The decision to position the policy in this way was underscored by the potential for prejudice or discrimination due to its focus on the menstrual cycle. Accommodations for menstruation and menopause have since been removed from the policy “to make it more palatable” [48]. 

These policy outcomes point to the salience of a ‘gender sensitive approach attuned to occupational health and safety,’ so that provision and uptake might be supported. It might be the case that industry- or profession-specific policies strike a good balance in terms of inclusiveness and levels of specificity. General laws might not be able to address the dynamics of all industries and professions, whereas employer-specific policies might be too embedded in or have emerged from a specific workplace culture and are not transferrable. Moreover, trade unions might have a stronger rights focus (i.e., utilising the rationale of gender occupational health concerns) and therefore may more meaningfully engage women in the design or decision-making process as compared to a top-down approach in employer-specific policies [81]. 

### 4.2. Disparity in the Literature around Menstrual Disorders and Work

The lack of evidence-based inquiry focused on addressing menstrual disorders at work remains problematic [6,11,12]. Over 70% of women and girls under the age of 25 report regular dysmenorrhea, with increased pain strongly correlating with increased absenteeism and reduced performance in school, university, and/or early career [87,88]. Dysmenorrhea and menstrual disorders profoundly impact some women and girls’ ability to fully participate in education and work, and contribute to cumulative economic disadvantage and financial burden [15]. Therefore, addressing the symptom management of these issues at work could improve absenteeism and presenteeism and alleviate the burden of illness for individuals. 

### 4.3. Gendered Language

Most articles (85%) use the term ‘women’ when referring to menstruation, menstrual disorders, and menopause at work. No articles explicitly include or seek to understand the experiences of trans men, intersex, non-binary, and gender diverse people who may also menstruate, have menstrual disorders, and/or are experiencing menopause at work. There is growing literature in health, humanities and law which recognises the health, political and social experiences of people other than non-cis women who menstruate [89,90]. Therefore, the lack of literature on trans, intersex, non-binary, and gender diverse people in the workplace context is significant.

### 4.4. Design, Implementation, and Evaluation Gaps

Substantial gaps in reporting exist around the design process, implementation plan, and evaluation of interventions, with close to one third reporting how interventions were designed. For most articles, there is uncertainty about how or why policies or practices were developed and established. Even fewer articles provided guidance on or the details of the intervention evaluation undertaken. This ambiguity means that knowledge of uptake, implementation, and evaluation of practices is limited, which creates difficulties for informing evidence-based implementation of interventions that may benefit women at work. 

To provide further nuanced insights, we have categorised the rationales underpinning/driving the interventions (pronatalist, economic rationalism, gendered occupational health concern, cultural shift to achieve gender equity in the workplace, and reducing shame and stigma). Interventions must be evaluated within their context and setting because the way policy is problematized constructs both the issue and the resultant impact, as well as the possibility for the policy to contribute to broader equity and empowerment at work [91,92]. Therefore, we need to understand the rationale(s) driving the intervention to support better design, implementation, and evaluation in the future [1]. 

Future policies should include an appropriate design process, implementation, and evaluation plan. Furthermore, employers should detail the context or rationale driving the intervention. This will help in the evaluation of the policy so researchers and relevant stakeholders can understand the factors that influence its efficacy and gain insight into its potential for translation to other settings. 

## 5. Strengths and Limitations

This review provides a comprehensive overview and analysis of the current published work across academic and grey literature on an under-researched topic area—interventions to reduce the impact of menstruation and menopause at work. This study is of use to academics and practitioners alike who are interested in workplace gender equity and interventions for menstrual and menopause equity in the workplace. Not only does this paper inform readers about the types of interventions that have been implemented, but it also provides insights into the motivation behind the intervention, the type of intervention, and any available information about how it was evaluated. The primary strength of this review is the use of ten databases that cover a potential range of topics, from biomedical interventions to workplace policies and legal opinions. To ensure that the literature had been extensively mapped, we supplemented this approach through reviews from topic experts. Additionally, the use of the JBI structure for the scoping review ensures that the broad range of cross-disciplinary information is presented in a systematic way. There are two major limitations of this scoping review. Firstly, only articles published in English were included. Secondly, our scoping review did not assess the underlying quality of the studies included. This is common as scoping reviews are designed to provide an evidence map, irrespective of study quality, and given the wide range of materials included, from policy documents to more formal academic material, no standard measure for the quality of this information exists [21]. 

## 6. Conclusions

This scoping review contributes to the literature by synthesising existing interventions designed to support women managing menstruation, menstrual disorders, and menopause in the workplace. While there is a growing body of literature addressing menopause in the workplace, the problem to be solved is the lack of interventions focusing on women’s experiences or support needs related to menstruation and/or menstrual disorders in the workplace. Future studies could address several research gaps identified in this scoping review. Firstly, there is a need for more research on the different types and effectiveness of interventions to support women’s menstrual health in the workplace. Secondly, there is a lack of literature on trans, intersex, non-binary, and gender diverse individuals in the workplace context, which is a significant limitation. Third, more research is needed on the design process, implementation plan, and evaluation of interventions to guide future policies and practices. Lastly, exploring the economic and social impact of menstrual disorders and menopause on women in the workplace would provide valuable insights. 

Based on the evidence obtained in this review, there is a strong case for policy and practice change to support women’s menstruation, menstrual disorders, and transition through menopause in the workplace. Several gaps in policies, guidelines, and practices have been identified, highlighting the need for interventions to promote gender equity in the workplace. Policies should be developed to ensure adequate physical work environments, flexible work arrangements, education and awareness training, and access to reasonable accommodations for menstrual pain and symptoms. Additionally, employers should foster a supportive workplace culture that encourages open communication and informed understanding regarding menstruation, menstrual disorders, and menopause. Further research on the effectiveness of interventions is necessary to inform future policies and practices. Overall, this scoping review presents a compelling case for policy and practice development/change to support women menstruating, those with menstrual disorders, or those who are transitioning through menopause in the workplace.

## Figures and Tables

**Figure 1 healthcare-11-02945-f001:**
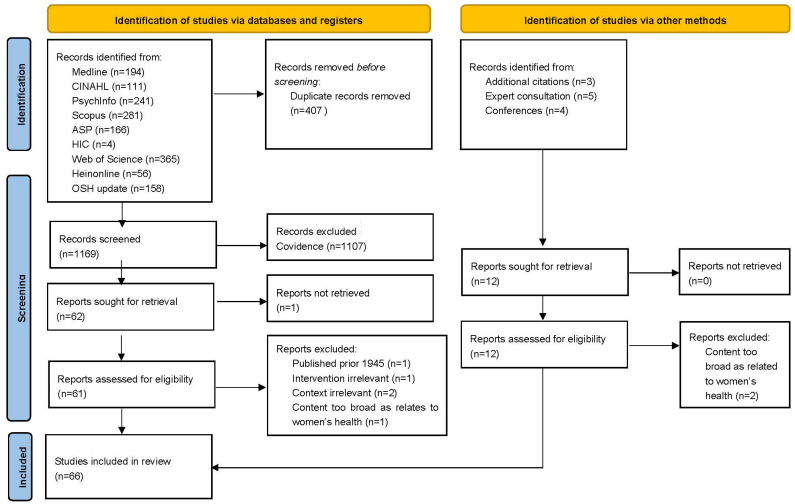
PRISMA 2020 flow diagram. Adapted from: Ref. [25]. For more information, visit: http://www.prismastatement.org/.

**Figure 2 healthcare-11-02945-f002:**
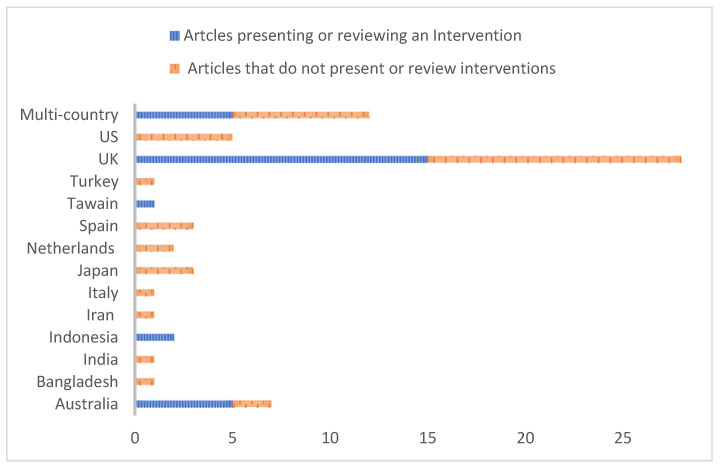
Number of articles that present or do not present an intervention, by geographic location.

**Table 1 healthcare-11-02945-t001:** Search terms, combinations, and database application.

Database		Search Strategy				
		Concept 1: Workplace	AND	Concept 2: Intervention	AND	Concept 3: Women’s Health
Medline (Ebsco) n = 193	Subject heading	MeSH workplace MeSH occupational health		MeSH Policy exp MeSH Jurisprudence		MeSH Menstruation MeSH Menstruation disturbances exp MeSH Menopause exp MeSH Endometriosis MeSH Polycystic ovary syndrome
	Text words	Workp * Works *		Guideline * Intervention * Program *		Menstrua * Menopaus * Dysmenorrh *
CINAHL (Ebsco) n = 111	Subject heading	MeSH work environment MeSH occupational health		MeSH Legislation		MeSH Menstruation MeSH Menstruation disturbances exp MeSH Menopause exp MeSH Endometriosis MeSH Polycystic ovary syndrome
	Text words	Workp * Works *		Polic * Law * Guideline * Intervention * Program *		Menstrua * Menopaus * Dysmenorrh *
APA PsychINFO (Ebsco) n = 241	Subject Heading			MM Laws MM Workplace intervention MM Intervention		MM Menstruation MM Menstrual disorders exp MM Menopause
	Text words	Work * Occupation * Employe *		Polic * Guideline * Program *		Menstrua * Menopaus * Dysmenorrh * Polycystic ovar * Endometriosis
Academic Search Premier (Ebsco) n = 166	Key Words	Workplace * Works * Work environment		Polic * Law * Guideline * Program * Intervention *		Menstrua * “Menstrua * disorder*” Menopaus * “Polycystic ovar *” Dysmenorrh *
	Subject terms	DE office environment DE work environment		DE workforce planning DE guidelines		DE menstruation DE endometriosis DE menstruation disorder exp DE menopause DE polycystic ovary syndrome
Humanities International Complete (Ebsco) n = 4	Key words	Workplace Works *		Polic * Law * Guideline * Program * Intervention *		Menstrua * “Menstrua * disorder *” Menopaus * “Polycystic ovar *” Dysmenorrh * Endometriosis
	Subject terms					DE menstruation

* Refers to truncated word roots to capture multiple derivations, e.g., menstrua* will capture menstruation, menstruator, menstruating, etc. MeSH refers to MeSH terms. ‘MM’ searches the exact subject heading. DE refers to Heading or Keyword.

**Table 2 healthcare-11-02945-t002:** Search strings and database application.

Database	Search String
**Scopus** **n = 281**	(TITLE-ABS-KEY (workplace OR worksite * OR works *) AND TITLE-ABS-KEY (polic * OR law * OR guideline * OR program * OR intervention *) AND TITLE-ABS-KEY (menstrua * OR menopaus * OR endometriosis OR “polycystic ovar *” OR dysmenorrh *))
**Web of Science** **n = 365**	((TS = (workplace OR worksite OR worker *)) AND TS = (polic * OR law * OR intervention * OR program *)) AND TS = (menstr * OR endometriosis OR “polycystic ovar *” OR menopaus * OR dysmenorrh *)
**Heinonline** **n = 56**	“Workplace policy menstruation”~200 OR “workplace law menstruation”~200 OR “workplace legislation menstruation”~200 OR “workplace regulation menstruation”~200 OR “workplace statute menstruation”~200 OR “workplace policy endometriosis”~200 OR “workplace policy polycystic ovary”~200 OR “workplace policy menopause”~200 OR “workplace policy dysmenorrhea”~200
**OSH Update** **n = 158**	((TS = (workplace OR worksite OR worker *)) AND TS = (polic * OR law * OR intervention * OR program *)) AND TS = (menstr * OR endometriosis OR “polycystic ovar *” OR menopaus * OR dysmenorrh *)

* Refers to truncated word roots to capture multiple derivations, e.g., menstrua* will capture menstruation, menstruator, menstruating, etc.

**Table 3 healthcare-11-02945-t003:** Inclusion criteria for scoping review.

	Inclusion Criteria
Population	16 years or older
Concept	Interventions such as policy, guidelines, or laws. Women’s health relating to: menopause or disorders of the menstrual cycle (e.g., endometriosis, dysmenorrhea, polycystic ovary syndrome). Comparison: any intervention with or without a comparator or evaluation
Context	Based in any workplace or workplace occupational health

**Table 4 healthcare-11-02945-t004:** Overview of included articles.

**Geographic Representation Included in Articles**	**N**
Australia	7
Bangladesh	1
India	1
Indonesia	2
Italy	1
Iran	1
Japan	3
Netherlands	2
Spain	1
Taiwan	1
Turkey	1
United Kingdom	28
United States	5
**Multi-country articles**	12
Australia, Austria, Belgium, Brazil, Bulgaria, Canada, China, Croatia, Cyprus, Czech Republic, Denmark, Ecuador, Egypt, Estonia, Finland, France, Germany, Greece, Hungary, India, Indonesia, Ireland, Israel, Italy, Japan, Latvia, Lithuania, Luxembourg, Malta, Netherlands, Nepal, New Zealand, Philippines, Poland, Portugal, Romania, Russia, Slovakia, Slovenia, South Korea, Spain, Sweden, Switzerland, Taiwan, Turkey, United Kingdom, Vietnam, Zambia.	
**Type of article**	
Narrative review	15
Systematic review	3
Editorial	4
Empirical study:	
Observational cross-sectional (mixed methods)	11
Observational cross-sectional (qualitative)	5
Descriptive cross-sectional (qualitative)	1
Randomised control trial (mixed methods)	2
Retrospective cohort (mixed methods)	1
Descriptive case study (qualitative)	1
Interventional (mixed methods)	2
Interventional (qualitative)	3
Grey literature:	
Research report	2
Working paper	1
Trade union, federations, or industry report	10
Guidelines or assessment tool	4
Policy template	1
Article/blog	2
**Presents or reviews an intervention (policy, guidelines, or laws)?**	
Yes	28
No	38

**Table 5 healthcare-11-02945-t005:** Categorization of articles that present or do not present/review an intervention.

Type of Article	Articles That Present or Review an Intervention (n)	Articles That Do Not Present of Review and Intervention (n)	Total (N)
Narrative Review	8	5	12
Systematic Review	2	1	3
Editorial	1	3	4
Study	3	23	26
Grey Literature	14	6	20
Total (N)	28	38	66

**Table 6 healthcare-11-02945-t006:** Comprehensive coverage of 24 articles presenting or reviewing an intervention.

Author	Year	Ref	Country	Target Population ^1^	Intervention Focus	Workforce	Type of Article
DuBrin, A. and Bizo, M	1988	[32]	Australia	(cis) women with premenstrual changes	Menopause	Unspecified	Narrative review
Golding, G. and Hvala, T.	2021	[33]	Australia	Menstruating (cis) women	Menstruation	Unspecified	Narrative review
Safe Work Australia	2019	[34]	Australia	Individuals with endometriosis	Endometriosis	Unspecified	Grey literature: Guidelines
Melican, C. and Mountford, G.	2016	[35]	Australia	Menstruating and menopausal employees	Menstruation and menopause	Unspecified	Grey literature: Policy Template
Carter, S. et al.	2021	[3]	Australia	Menstruators	Menopause	Unspecified	Journal editorial
Kridasaksana, D. et al.	2020	[36]	Indonesia	Menstruating (cis) women	Menstruation	Unspecified	Narrative review
Lahiri-Dutt, K. and Robinson, K.	2008	[37]	Indonesia	(cis) Women	Menstruation	Mining industry	Descriptive case study
Chang, C. et al.	2011	[38]	Taiwan	Menstruating (cis) women	Menstruation	Unspecified	Observational retrospective cross-sectional study—mixed methods
Dudley, C. et al.	2017	[27]	UK	(cis) Women	Endometriosis	Unspecified	Narrative review
Griffiths, A. et al.	2016	[39]	UK	Menopausal (cis) women	Menopause	Unspecified	Narrative review
ACAS	2022	[40]	UK	Employers and staff	Menopause	Unspecified	Grey literature: Guidelines
Targett, R. and Beck, V.	2022	[41]	UK	Employees of local council	Menopause	Local government	Observational cross-sectional—mixed methods
Dean, E.	2018	[42]	UK	Menopausal (cis) women	Menopause	Nursing	Grey Literature: Industry report
Noble, N.	2021	[43]	UK	Menopausal (cis) women	Menopause	Nursing	Narrative review
Hardy, C. et al.	2018	[2]	UK	Menopausal (cis) women	Menopause	Trade unions and federations	Systematic review
HAZARDS	2003	[28]	UK	Menopausal (cis) women	Menopause	Trade unions and federations	Grey literature: Industry report
International Union of Food (IUF)	2019	[44]	UK	(cis) Women	Menopause	Trade unions and federations	Grey literature: Industry report
The Royal Society for the Prevention of Accidents (RoSpa)	2020	[29]	UK	Menopausal (cis) women	Menopause	Trade unions and federations	Grey literature: Industry report
Trades Union Congress (TUC)	2008	[45]	UK	(cis) Women	Menopause	Trade unions and federations	Grey literature: Checklist template
TUC	2013	[30]	UK	Menopausal women	Menopause	Trade unions and federations	Grey literature: Guidelines
Unite the Union	2012	[31]	UK	(cis) Women	Menopause	Trade unions and federations	Grey literature: Industry report
Grohs, B. and Harriss, A.	2019	[46]	UK	Menopausal (cis) women	Menopause	Thermally challenging work environments	Grey literature: Industry report
European Agency for Safety and Health at Work (EU-OSHA)	2014	[47]	Multi-country	(cis) Women	Menopause	Trade unions and federations	Grey literature: Industry reports
Jack, G. et al.	2016	[18]	Multi-country	Menopausal (cis) women (age 45 to <60)	Menopause	Trade unions and federations	Systematic review
Baird, M. et al.	2021	[1]	Multi-country	People who menstruate	Menstruation	Varies	Systematic review
Bennet, J. et al.	2021	[48]	Multi-country	Menstruating (cis) women	Menstruation	Unspecified	Grey literature: Working paper
Rees, M. et al.	2021	[49]	Multi-country	Menopausal (cis) women	Menopause	Unspecified	Narrative review
Yoeli, H. et al.	2021	[50]	Multi-country	Menopausal (cis) women	Menopause	Working in casual contracts/informal or insecure jobs	Narrative review

^1^ Target population reflects the gendered language used throughout report, e.g., some papers used the term (cis) women, or some use ‘individuals’, ‘employees’, etc.

**Table 7 healthcare-11-02945-t007:** Type of recommendations from articles presenting or reviewing interventions.

Recommendations	N (Number of Articles with Recommendation)	% (N/28)
Physical work environment		
Promoting adequate fitting and flexible wearing of PersonalProtective Equipment (PPE) and uniforms	6	21%
Adequate/increased lighting	3	11%
Access to cold drinking water and/or food (e.g., vending machines)	8	29%
Accessible and clean toilets	7	25%
Additional and flexible breaks and rest areas	7	25%
Adequate ventilation & temperature control (e.g., desk fans, aircon, access to/near windows)	7	25%
Information and training		
Promoting and conducting awareness raising campaigns focused on gendered health	14	50%
Gender-sensitive or gendered health and safety checklists	8	29%
Guidelines/brochures for leaders, managers, and staff	7	25%
Counselling services and/or education for employees to help support their health needs	5	18%
Policy and processes		
Flexible hours		
Flexible shift patterns	5	18%
Flextime (e.g., offer to work from home—varying hours and workdays)	8	29%
Additional Leave		
Additional paid leave (no medical certificate required)	9	32%
Additional paid leave (requires medical certificate)	2	7%
Structured unpaid leave (set # days, requires medical certificate)	3	11%
Policy		
Recognition of reproductive and gynaecological health in workplace policy	5	18%
Renewing occupational health and safety frameworks	6	21%
Integrating education and gender into organisations’ planning, administration, and daily working practises	8	29%
Processes		
Pathway for female staff to access confidential disclosure	7	25%
Ensure women’s participating in policy discussion	3	11%
Workplace support / wellbeing champion (e.g., support from occupational health, human resources, trade union representatives, GPs, line managers, counsellors, Employee Assistance Programmes, peers, welfare officers)	4	14%

**Table 8 healthcare-11-02945-t008:** Intervention implementation and evaluation.

**Implementation**	**N**
Consistent public-sector implementation (i.e., set leave entitlements across industries/employers)	2
Inconsistent public sector implementation (e.g., leave entitlements up to discretion of employer)	2
Organisation implements top-down: paid leave and/or flexitime system	2
Inconsistent industry, trade unions and federations implementation (e.g., leave, accommodations, counselling services are up to the discretion of each organisations/employer /employee)	17
Unclear joint-system implementation—provides recommendations but up to organisations in each system to implement and work together	3
Multi-intervention review (with multiple implementations)	2
Consistent public-sector implementation (8) Inconsistent public sector implementation (5) Private sector implementation paid leave and/or flexitime system (15) Private sector implementation code that refences menstrual leave to the national legislative regimes of countries in which it operates (1) Bottom-up (employee-led initiatives)	
**Evaluation**	
Ongoing monitoring and evaluation strategy (includes tools, timelines, and multiple approaches)	1
Participatory evaluation approach: focus groups and case studies	2
Evaluation from a health law perspective	1
Not available	24

**Table 9 healthcare-11-02945-t009:** Overview of all articles included in scoping review.

Geographic Representation	N	Workforce	n	Presents or Reviews an Intervention	n	Rationale Driving Intervention (n)	Article Type (n)	References
Australia	7	Unspecified	5	Yes	5	Economic Rationalism (4); Culture Shift (1)	Narrative Review (2) Grey literature: policy template (1) Grey literature: Guidelines (1) Journal Editorial (1)	[3,32,33,34,35]
		Healthcare and Higher education sectors	2	No	2	-	Observational cross-sectional—mixed methods (2)	[18,52]
Bangladesh	1	Garment sector	1	No	1	-	Observational cross-sectional—mixed methods (1)	[53]
India	1	Textile Industry	1	No	1	-	Interventional Study—qualitative (1)	[54]
Indonesia	2	Unspecified	1	Yes	1	Pronatalist (1)	Narrative Review (1)	[36]
		Mining industry	1	Yes	1	Pronatalist (1)	Descriptive Case study (1)	[37]
Iran	1	Unspecified	1	No	1	-	Systematic review (1)	[19]
Italy	1	Administrative officers in public sector	1	No	1	-	Observational cross-sectional—mixed methods	[55]
Japan	3	Unspecified	2	No	2	-	Interventional Study (qualitative) (1) Randomised control trial (1)	[56,57]
		Journalists/Print media	1	No	1	-	Interventional Study (qualitative)	[58]
Netherlands	2	Unspecified	1	No	1	-	Interventional Study (mixed methods)	[59]
		Low-paid roles at university medical centre	1	No	1	-	Observational cross-sectional—mixed methods	[60]
Spain	1	Working outside of home	1	No	1	-	Observational cross-sectional—mixed methods	[61]
Taiwan	1	Unspecified	1	Yes	1	Pronatalist (1)	Observational retrospective cross-sectional study—mixed methods	[38]
Turkey	1	Working in Universities	1	No	1	-	Observational cross-sectional—mixed methods	[26]
United Kingdom	28	Unspecified	10	Yes	3	Culture Shift (2) Gendered Occupational Health Concern (1)	Narrative Review (2); Grey literature: Guidelines (1)	[27,39,40]
				No	7	-	Grey literature: article/blog (1) Journal Editorial (1) Interventional (mixed methods) (1) Narrative Review (1) Observational cross-sectional (qualitative) (1) Descriptive cross-sectional-qualitative)\(1) Randomised control trial (1)	[62,63,64,65,66,67,68]
		Ambulance staff	1	No	1	-	Observational cross-sectional—mixed methods (1)	[69]
		Local Councils (all staff)	1	Yes	1	Culture Shift (1)	Observational cross-sectional—mixed methods	[41]
		Non-manual roles	1	No	1	-	Observational cross-sectional—mixed methods	[70]
		Nursing	3	Yes	2	Economic Rationalism (2)	Grey Literature: Industry report (1) Narrative review (1)	[42,43]
				No	1	-	Grey Literature: research report	[71]
		Police	1	No	1	-	Observational cross-sectional—qualitative (1)	[72]
		Office staff in a social enterprise	1	Yes	1	Culture Shift (1)	Narrative Review	[73]
		Teaching staff in faculty of medicine	1	No	1	-	Observational cross-sectional—mixed methods	[74]
		Trade unions and Federations	8	Yes	7	Gendered Occupational Health Concern (7)	Grey Literature: checklist template (1) Grey Lit: trade union & federation report (4) Grey lit: guidelines (1) Systematic review (1)	[2,28,29,30,31,44,45]
				No	1	-	Editorial (1)	[75]
		Working in thermally challenging environments	1	Yes	1	Gendered Occupational Health Concern (1)	Grey literature: industry report (1)	[46]
United States	5	Unspecified	5	No	5	-	Observational cross-sectional—qualitative (2) Narrative review (3)	[76,77,78,79,80]
Multi-country	12	Trade unions and federations	4	Yes	2	Gendered occupational health concern (2)	Grey Literature: Union & Federation report (2)	[18,47]
				No	2	-	Grey Literature: Union & Federation report (2)	[51,81]
		Varies/unspecified	7	Yes	3	Pronatalist Economic rationalism Reducing shame and stigma	Grey Literature: Working paper (1) Narrative Review (1) Systematic Review (1)	[1,48,49]
				No	4	-	Grey Literature: Research Report (1) Observational cross-sectional—qualitative (1) Editorial (1) Grey Literature: Article/blog	[8,82,83,84]
		Working in casual contracts/informal or insecure jobs	1	No	1	-	Narrative Review (1)	[50]

## Data Availability

Not applicable.

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
