# Peer review of "Policies, Guidelines, and Practices Supporting Women’s Menstruation, Menstrual Disorders and Menopause at Work: A Critical Global Scoping Review"

_healthcare, 2023, doi:10.3390/healthcare11222945_

Round 1

Reviewer 1 Report

Comments and Suggestions for Authors

Dear editor:

Thanks for the opportunity to review the manuscript. The article has an interesting topic, but the data presentation is not done well. I think that discussing multiple subjects and topics in this article may have led to scattered results and a lack of comprehensive coverage of the subject. It would have been more effective to address each subject separately and provide more detailed analysis. By the way, it is important to note that the review did not evaluate the underlying quality of the studies included.

Please see my feedback and suggestions:

1.      As far as I know, searching in EMBASE is crucial for finding studies from eastern countries.

2.      Did you search the keywords in MESH or Emtree? The search terms should be found in both Embase (Emtree) and PubMed (Medical Subject Headings/MeSH), and a combination of these terms should be used to develop an appropriate electronic search strategy.

3.      It would be helpful to introduce every acronym before using it in the text (e.g., PPE).

4.      Could you please provide information on which studies were excluded?

5.      I would like to request the classification of Table 6, which pertains to recommendations based on interventions in menstruation, menstrual disorders, and menopause.

6.      I would appreciate if you could include the strengths of your study.

7.      In my opinion, the problem to be solved is the lack of policies, guidelines, and practices supporting women's menstrual health in the workplace. Please add this in the conclusion.

8.      My suggestion for further studies would be to address several research gaps identified in this scoping review. Firstly, there is a need for more research on the effectiveness of interventions to support women's menstrual health in the workplace. Secondly, there is a lack of literature on trans, intersex, non-binary, and gender diverse individuals in the workplace context, which is a significant limitation. Thirdly, more research is needed on the design process, implementation plan, and evaluation of interventions to guide future policies and practices. Lastly, exploring the economic and social impact of menstrual disorders and menopause on women in the workplace would provide valuable insights. Please add this in the conclusion.

9.      Based on the evidence obtained in this review, there is a strong case for policy and practice change to support women's menstrual health in the workplace. Several gaps in policies, guidelines, and practices have been identified, highlighting the need for action to promote gender equity and social justice. Policies should be developed to ensure access to menstrual products, flexible work arrangements, and accommodations for menstrual pain and symptoms. Additionally, employers should foster a supportive workplace culture that encourages open communication and understanding regarding menstrual health. Further research on the effectiveness of interventions is necessary to inform future policies and practices. Overall, this scoping review presents a compelling case for policy and practice change to support women's menstrual health in the workplace. Please add this in the conclusion.

10.  Based on the findings, my suggestion for healthcare providers is to be aware of the impact of menstrual disorders and menopause on women in the workplace and provide appropriate support and treatment. Policy makers should develop and implement policies and guidelines that support women's menstrual health in the workplace, including access to menstrual products, flexible work arrangements, and accommodations for menstrual pain and symptoms. Employers should create a supportive workplace culture that promotes open communication and understanding of menstrual health. Additionally, further research on the effectiveness of interventions is needed to guide future policies and practices. Please add this in the conclusion.

Comments on the Quality of English Language

Minor corrections is needed.

Author Response

Please see the attachment for Reviewer 1 

Reviewer 2 Report

Comments and Suggestions for Authors

The abstract should conclude with a sentence on the overall theoretical and practical implications of the results.

It is not fully clear to me what exactly the contribution is that is being claimed in this article?  Is it simply a replication of previous studies, or does it solve an open issue in the rather large body of prior literature on learning segmenting principle. Moreover, the Introduction is very short and does really lay a good foundation for the context of the study.

Instead of setting their aim in the frame of a simple question, I would recommend that the authors attempt to present the key objectives of their study with regards to what is presently known (i.e. literature), thus highlighting the added value of the article.

Please add the research hypothesis (as they are not with backed with theoretical considerations)

"Consider referencing the 'Web of Science' database. It's a rich source of research in your domain.

"It's not enough to merely state that the article included 64 articles. These should be displayed in a comprehensive table, showcasing the categories mentioned in the article: authors, year, country, gender, population, sample size (N), workforce, type of article

The article citations and the reference list need to be CAREFULLY examined! There are a number of places (I won’t list them all) where the citation in the text is not included in the reference list, or the year is wrong, or a page number is missing for a quote, or the name is wrong

Comments on the Quality of English Language

N

Author Response

Please see attachment for reviewer 2

Round 2

Reviewer 1 Report

Comments and Suggestions for Authors

I think that the authors did a great job addressing review comments. Anyhow, the authors didn’t submit a point-by-point reply to the comments.

Reviewer 2 Report

Comments and Suggestions for Authors

Paper  is now suitable for publication

Comments on the Quality of English Language

N/R